# Rare and Common Variants Uncover the Role of the Atria in Coarctation of the Aorta

**DOI:** 10.3390/genes13040636

**Published:** 2022-04-02

**Authors:** Wenjuan Zhu, Kylia Williams, Cullen Young, Jiaunn-Huey Lin, Polakit Teekakirikul, Cecilia W. Lo

**Affiliations:** 1Centre for Cardiovascular Genomics and Medicine, Chinese University of Hong Kong, Hong Kong 999077, China; wenjuan.zhu@link.cuhk.edu.hk (W.Z.); pteekakirikul@gmail.com (P.T.); 2Department of Developmental Biology, University of Pittsburgh School of Medicine, 530 45th St, Pittsburgh, PA 15201, USA; kyliaawilliams@gmail.com (K.W.); ycullen@pitt.edu (C.Y.); 3Department of Critical Care Medicine, University of Pittsburgh School of Medicine, 530 45th St, Pittsburgh, PA 15201, USA; jiuannhuey.lin5@upmc.edu

**Keywords:** *MYH6*, coarctation of the aorta, *PCHDA*, atrial dominance

## Abstract

Coarctation of the aorta (CoA) and bicuspid aortic valve (BAV) often cooccur and are genetically linked congenital heart defects (CHD). While CoA is thought to have a hemodynamic origin from ventricular dysfunction, we provide evidence pointing to atrial hemodynamics based on investigating the genetic etiology of CoA. Previous studies have shown a rare *MYH6* variant in an Icelandic cohort, and two common deletions in the protocadherin α cluster (*PCDHA* delCNVs) are significantly associated with CoA and BAV. Here, analysis of a non-Icelandic white CHD cohort (*n* = 166) recovered rare *MYH6* variants in 10.9% of CoA and 32.7% of BAV/CoA patients, yielding odds ratios of 18.6 (*p* = 2.5 × 10^−7^) and 20.5 (*p* = 7.4 × 10^−5^) for the respective association of *MYH6* variants with CoA and BAV/CoA. In combination with the *PCHDA* delCNVs, they accounted for a third of CoA cases. Gene expression datasets for the human and mouse embryonic heart showed that both genes are predominantly expressed in the atria, not the ventricle. Moreover, cis-eQTLs analysis showed the *PCHDA* delCNV is associated with reduced atrial expression of *PCHDA10*, a gene in the delCNV interval. Together, these findings showed that *PCDHA/MYH6* variants account for a substantial fraction of CoA cases. An atrial rather than ventricular hemodynamic model for CoA is indicated, consistent with the known early atrial functional dominance of the human embryonic heart.

## 1. Introduction

Development of the cardiovascular system is regulated by blood flow, with hemodynamic perturbation likely contributing to various congenital heart defects (CHD), such as coarctation of the aorta (CoA), bicuspid aortic valve (BAV), or other related CHD with left ventricular outflow tract obstructions (LVOTO) [1,2,3,4]. CoA is characterized by narrowing or constriction of the proximal portion of the descending aorta. It accounts for 5–8% of CHD [5] and is suggested to arise from ventricular dysfunction causing reduced aortic blood flow [6,7]. Severe CoA cases usually present neonatally with cardiogenic shock and heart failure, but milder cases may remain undiscovered until hypertension emerges. In 50% of CoA cases, BAV cooccurs with aortic valve comprised of two rather than three valve cusps. BAV is one of the most common birth defects with 1–2% prevalence [8], but often it remains undiscovered until aortic valve disease causes hemodynamic disturbance late in adult life.

A genetic etiology for CoA, BAV, and other LVOTO lesions are well described [9], and is evident from their high heritability and high recurrence risk [10]. As other LVOTO lesions, such as hypoplastic left heart syndrome (HLHS), can also have CoA and/or BAV, and isolated CoA and/or BAV can be observed in family members of HLHS index cases, this suggests genetic overlap for a broad spectrum of LVOTO lesions. A recent study identified a rare *MYH6* founder mutation (p.Arg721Trp) associated with CoA and also BAV, and was reported to account for 20% of all CoA cases in the Icelandic population [11]. However, the contribution of *MYH6* variants to CoA in other populations has not been investigated, although rare pathogenic *MYH6* variants have been reported in a family with HLHS [12]. As we recently also uncovered two common deletion copy number variants in the protocadherin α gene cluster (*PCDHA* delCNV) associated with the pathogenesis of LVOTO lesions [13], here we further investigated the combined role of common and rare variants in the genetic architecture of CoA and BAV. This study was carried out in a non-Icelandic cohort of LVOTO patients. We interrogated for the cooccurrence of rare and common variants in *MYH6/PCDHA* and their combined impact on CoA and BAV. We further examined the temporospatial profile of *MYH6* and *PCDHA* gene expression in the human heart, with further validation using cis-eQTL analysis tracked by the *PCHDA* delCNV.

## 2. Materials and Methods

### 2.1. Human Study Participants

One hundred sixty-six subjects with LVOTO were recruited with informed consent from Children’s Hospital of Pittsburgh (CHP) as previously described [13]. Personal identifiers of the study participants were encrypted in accordance with approved guidelines and regulations. All CHD diagnoses were clinically obtained based on echocardiography or MRI phenotyping. LVOTO phenotypes are defined as previously described and included isolated or complex bicuspid aortic valve (BAV), coarctation of the aorta (CoA), hypoplastic left heart syndrome (HLHS), and other LVOTO lesions [13].

### 2.2. Recovery of Rare Predicted Pathogenic MYH6 Variants

Whole exome sequencing (WES) data from the CHP patients was processed as previously described and high-quality *MYH6* variants were recovered [13]. Only protein-altering variants were retained, including predicted loss-of-function (LoF) mutations (nonsense, canonical splice-site, frameshift indels, and start loss), inframe indels, and predicted damaging missense mutation (D-Mis). As many missense variants are tolerant and would affect degree of enrichment of pathogenic variants in a case cohort, only predicted D-Mis called likely pathogenic by at least 4 of 8 prediction algorithms (SIFT, Polyphen2_HDIV, MutationTaster, MutationAssessor, FATHMM, PROVEAN, MetaSVM, M_CAP) were kept for downstream analyses.

For ensuring compatibility of the WES data with the GnomAD database, the following variant quality controls were included for recovery of rare pathogenic *MYH6* variants in the case cohort: (1) variant coverage is at >10x in >90% of samples; (2) SNV or indel genotype quality score ≥ 20 or 60, respectively, and read depth ≥ 10x; (3) minor allele frequency (MAF) of variants < 2 × 10^−4^ across all samples in gnomAD exome v2.1.1 [14]. This more stringent cutoff will allow comparison between different cohorts [15,16,17]. The 16.8 kb and 13.6 kb *PCDHA* delCNVs were recovered from the WES data as previously described [13].

### 2.3. Gene-Based Burden Testing

The Genome Aggregation Database (gnomAD) exome v2.1.1 (125,748 exomes) or ExAC (60,706 exomes) databases integrating large-scale exome sequencing projects with variable ancestry backgrounds were used as controls for gene burden analysis, similar to other published studies [16,17,18]. We tested whether there is significant excess of *MYH6* rare damaging variants in a case cohort with comparison to control cohort (gnomAD exome v2.1.1) using only high-confidence pathogenic variants as described above. For the control cohort (gnomAD exome v2.1.1), only *MYH6* variants with high-quality calls (PASS filter value) and with coverage at >10x in >90% of samples were retained for downstream analyses. The rare predicted pathogenic *MYH6* variants were extracted as described above. The total number of alleles evaluated in *MYH6* was taken as the median of the allele numbers recovered for all rare damaging *MYH6* variants as previously described [15,16]. Fisher’s exact test was used to estimate *p* value and the odds ratio (OR) with 95% confidence intervals (given 7 sub-phenotypes, *p* = 0.00714 for Bonferroni-corrected significance threshold) [15,17]. Similar burden analyses were conducted for rare synonymous variants in *MYH6,* which is not expected to be disease-related. This showed no significant increase in burden in cases vs. controls (*p* = 0.49).

### 2.4. Identification of TagSNPs for the 16.8 kb PCDHA delCNV in European-American Population

The reference panel from the 1000 Genomes Project (1KG) phase 3 [19,20] was applied to identify potential tagSNPs for the common 16.8 kb *PCDHA* delCNV (esv3606958 in 1KG) in a European-American population. Samples from a population with non-Finnish European or American ancestry (population code: EUR and AMR, *n* = 751) were extracted from the reference panel. A measure of LD (*D*′) and *r*^2^ was calculated using the Haploview 4.2 software (-maxdistance 500 -dprime -memory 5000) [21]. SNPs in complete linkage disequilibrium (LD) with the 16.8 kb delCNV (*r*^2^ = 1) were recovered as tagSNPs. This yielded four tagSNPs (rs77020201, rs76789733, rs115070123, and rs112910602) for the 16.8 kb *PCDHA* delCNV in the European-American population. TagSNPs for the 13.6kb delCNV were not found due to its rarity.

### 2.5. Determining eQTL Tissue Association Using Genotype-Tissue Expression (GTEx) Database

Data files were directly downloaded from the GTEx v8 portal [22,23] containing the variant-gene cis-eQTL associations tested to examine variant-gene associations in each tissue (*n* = 49) in the European-American subjects. The definition of eGenes in GTEx database refer to genes with at least one SNP in cis that is significantly associated with expression differences in that gene with a false discovery rate (FDR) of <0.05 [23]. cis-eQTL results of four tagSNPs were directly extracted from these downloaded data files.

### 2.6. Analysis of Publicly Available Single Cell RNA Sequencing Data

Publicly available raw single cell RNA sequencing (scRNAseq) data of the early mouse embryonic hearts from E8.5 to E10.5 [24] were downloaded from the NCBI GEO database under accession number GSE76118. The high degree of sequence homology between different genes in the protocadherin gene clusters causes mapping ambiguity. To address this multiple mapping issue, we used STAR [25] and RSEM [26] as described previously [27]. Briefly, reads were mapped to mouse (mm10) using STAR v2.7.0f_0328, allowing maximum 20 for multiple alignments (detailed parameters: --outSAMunmapped Within --outFilterType BySJout--outSAMattributes NH HI AS NM MD--outFilterMultimapNmax 20 --outFilterMismatchNoverReadLmax 0.06 --limitOutSJcollapsed 2,000,000 --limitIObufferSize 400,000,000) guided by parameters from the ENCODE project. Expression levels were quantified using RSEM v1.3.3. We generated a cell x gene transcripts per million (TPM) matrix at the gene level after aggregating expression of all cells together. Expression of all genes in the *Pcdha* cluster were combined to generate the expression of the entire *Pcdha* cluster. Downstream analyses were performed as described previously [24].

### 2.7. Additional Information on Data Presented in This Study

The flow diagram of bioinformatic analyses involved in this study was found in Appendix A.

## 3. Statistics

Statistical analyses were performed using the R package. Gene-based burden analysis that test the excess of *MYH6* rare damaging variants in LVOTO cohorts was statistically assessed using Fisher’s exact test, and odds ratio (OR) with 95% confidence interval (CI) was calculated. *p* values < 0.00714 (Bonferroni corrected) were considered statistically significant.

## 4. Results

### 4.1. Significant Association of Rare MYH6 Variants with CoA and BAV

A previous study showed association of a rare missense mutation p.Arg721Trp in *MYH6* with CoA and BAV in the Icelandic population [11]. To investigate the generalizability of this finding, we assessed for rare damaging *MYH6* variants in the whole exome sequencing data of a cohort of 166 white patients recruited in Pittsburgh with CoA, BAV, and other related LVOTO CHD. Eleven rare potentially pathogenic *MYH6* variants were recovered in 10 LVOTO patients in the Pittsburgh LVOTO cohort (Figure 1A, Appendix A). Comparison of the frequency of *MYH6* carriers in CoA vs. non-CoA subjects in the LVOTO cohort showed that *MYH6* variants were significantly enriched in subjects with CoA with *p* = 0.016 and OR of 5.19 (Appendix A). Further phenotype stratification showed *MYH6* variant frequency of 12.7% for CoA, 13.8% for BAV with CoA, and 6.02% for all LVOTO subjects combined. In comparison, examination of the GnomAD reference population yielded *MYH6* variant frequency of 0.776% (Appendix A).

Burden analysis showed enrichment with OR of 18.64 for CoA (*p* = 2.54 × 10^−7^), 20.47 for BAV/CoA (*p* = 7.43 × 10^−5^), and 8.20 for all LVOTO cases combined (*p* = 9.12 × 10^−7^) (Figure 1B, Appendix A). While complex CoA also yielded OR of 19.67 with *p* = 5.92 × 10^−3^, above the Bonferroni corrected *p* value of *p* = 0.00714, when GnomAD subjects of non-Finish European descent were used as controls, this became insignificant with *p* = 8.86 × 10^−3^, while other results were largely unchanged (Appendix A) Given that *MYH6* was previously implicated in HLHS, we also examined the HLHS patients in the LVOTO cohort and found no significant association of *MYH6* variants with HLHS (Appendix A). We noted that the *MYH6* variant identified in the Icelandic population, Arg721Trp, was not recovered in our LVOTO cohort, but a variant causing an amino acid substitution at the same position (Arg721Gln) was recovered in a patient with CoA and BAV (CoA/BAV) (Figure 1A, Appendix A). This patient was the only subject harboring two rare *MYH6* variants (Figure 1A). Overall, these findings replicated the association of *MYH6* variants with CoA and CoA/BAV in the Icelandic population, although the specific variant recovered in the Icelandic population was not observed.

### 4.2. Cooccurence of Rare MYH6 Variants with the PCDHA delCNV

Given the recent demonstration of the association of common *PCDHA* delCNVs with LVOTO, including CoA and BAV [13], we further investigated their co-occurrence with the MYH6 rare variants to assess the fraction of CoA cases that may be accounted for by the two variants, either alone or in combination. While the *MYH6* Arg721Trp mutation was observed in 20% of CoA cases in the Icelandic cohort [11], in the Pittsburgh cohort, rare *MYH6* variants accounted for only 10.9% of CoA cases, while *PCDHA* delCNV was found in 20% of CoA cases, and together they accounted for over 30% of all CoA cases (Figure 1C, Appendix A).

Among subjects with CoA and BAV, 13.8% had only the *MYH6* variant, 13.8% had only the *PCDHA* delCNV, and none had both. For complex CoA, 20% had the *PCDHA* delCNV vs. 13.3% for rare *MYH6* variants, and again, none had both (Figure 1C). It is notable that only one LVOTO patient showed co-occurrence of an *MYH6* variant with the *PCDHA* delCNV. No enrichment was observed when compared to the 1KG non-Finnish European control (*p* = 0.66; Appendix A). Together, these findings suggest the *MYH6* and *PCDHA* variants may act independently, impacting different tissues or different developmental processes important for development of the aorta/aortic valve.

### 4.3. MYH6 and PCDHA Expression in the Human and Mouse Heart

*MYH6* is known to be preferentially expressed in the human embryonic and adult atria [28]. This is also illustrated by spatial transcriptomic data of the developing human heart at 4.5–5, 6.5, and 9 weeks gestation [29], spanning the time of aorta and aortic valves morphogenesis (Figure 2A). *MYH6* transcript expression is observed predominantly in the atria during these early embryonic stages, consistent with previous study showing *MYH6* expression significantly declines in the ventricles with fetal age advancing from 47 to 110 days gestation [28]. Similarly, publicly available scRNAseq datasets of the early mouse embryonic heart show *Myh6* transcripts predominantly in the atria in the E8.5 to 10.5 mouse heart, consistent with findings from human spatial transcriptomics data (Figure 2) [24].

Using the mouse scRNAseq data [24], we also investigated expression from the *Pchda* gene cluster using a modified bioinformatics pipeline (see Section 2). This analysis showed that the *Pcdha* gene cluster is expressed at low levels in the mouse embryonic heart tissue, with expression found mostly in the atria in the early embryo (Figure 2B), in line with our previous findings [27]. *PCHDA* transcript expression could not be investigated from available human embryonic spatial transcriptomics data given low sequencing depth [29]. Nevertheless, the combined mouse and human findings point to atrial dysfunction as likely the underlying cause for CoA associated with the *MYH6* and *PCDHA* mutations.

### 4.4. eQTL Shows PCHDA delCNV Associated with Reduced PCDHA10 Expression in Human Atria

To assess the impact of the *PCDHA* delCNV on transcription from the *PCDHA* gene cluster, we used the GTEx database to investigate eQTLs (expression Quantitative Trait Loci) linked to the delCNV in a wide range of human tissues, including the heart (aorta, atria, ventricles) [23]. As GTEx does not provide eQTL data for CNVs, we used four common tagSNPs (rs77020201, rs76789733, rs115070123, and rs112910602) recovered from the 1000 genome project (1KG; phase 3) to track the *PCDHA* delCNV [20]. These four tagSNPs are in complete linkage disequilibrium (LD) with the common 16.8 kb *PCDHA* delCNV spanning *PCDHA8-10* (r^2^ = 1) in 1KG non-Finnish European-American population (*n* = 751), indicating their utility for genotyping subjects for the 16.8 kb *PCDHA* delCNV. Three of the tagSNPs, which are intron variants in *PCDHA8-10*, were among the top 5 cis-eQTLs found, while tagSNP rs115070123 was absent in GTEx. The 13.6 kb *PCDHA* delCNV was not examined as no tagSNPs were available for this delCNV, likely due to its much lower prevalence.

Examination of the GTEX database for the three tagSNPs showed they had a strong negative effect on *PCDHA* expression, including atria tissue of the heart (Figure 3). Using tagSNP rs77020201 for more detailed eGene-eQTL analysis showed specific reduction in the expression of *PCDHA10* (slope = −0.593 and *p* = 3.8 × 10^−8^). This is observed for atrial tissue (Figure 3A,B), and other human tissues, including the brain, where protocadherins are known to be highly expressed (Figure 3A,B). Examining all tissues from European-American subjects in the GTEx database showed the reduced expression of *PCHDA8-10,* the three *PCDHA* genes in the 16.8 kb *PCDHA* delCNV interval, while genes flanking either side, *PCHDA5-7* and *11–12,* showed elevated expression, indicating possible compensatory increase in expression (Figure 3C). These findings are in agreement with observations in the *Pcdha* knockout mice that similarly showed increased expression of *Pcdha* genes flanking either side of deletions in the *Pcdha* gene cluster [30]. Overall, these findings confirmed expression of the *PCDHA* gene cluster in the human heart is atrial specific, with no expression detected in the ventricle.

## 5. Discussion

Our findings replicated the significant association of *MYH6* variants with CoA and BAV previously reported in the Icelandic population, confirming the role of rare pathogenic *MYH6* variants in malformation of the aorta and aortic valve. While we did not recover the Icelandic founder mutation p.Arg721Trp, another missense mutation was recovered in the same amino acid residue, supporting its functional importance. In contrast to 20% of CoA cases being explained by the *MYH6* variant in the Icelandic population, only 10.9% of CoA cases in the Pittsburgh cohort were associated with rare *MYH6* variants. However, rare *MYH6* variants together with the common *PCDHA* delCNVs were associated with nearly a third of CoA cases in the Pittsburgh cohort.

Our in silico analysis of transcriptomic data from the human and mouse embryonic heart showed *MYH6* and *PCDHA* are both expressed predominantly in the atria. This is supported by the eQTL analysis, which showed association of the 16.8 kb *PCDHA* delCNV with reduced expression of *PCDHA10*, gene in the deletion interval, in human atrial tissue. We note mice with mutation in *Pcdha9*, the mouse ortholog of human *PCDHA10,* can exhibit BAV [31]. Thus, mutation in either *MYH6*, a sarcomeric heavy chain myosin, or *PCDHA* mediating cell-cell adhesion, can cause CoA via expression in the atria. Thus, each gene is likely to have importance for force generation in the atria, rather than ventricle as suggested by the current ventricular hemodynamic model of CoA (Figure 4A).

It is likely that deficiency of either gene in the early embryonic heart may cause decreased atrial contractility with reduced blood flow that can lead to CoA (Figure 4B). This atrial hemodynamic model of CoA is well supported by previous studies showing atrial functional dominance in the early mouse and human embryonic heart [33,34]. This is likely a reflection of the higher ventricular preload causing cardiac output to be dependent on atrial rather than ventricular contractility [33,34]. With this new atrial model of CoA, we predict other genes regulating atrial contractility may contribute to the penetrance of the CoA phenotype (Figure 4). We note several studies reported circulating biomarkers are associated with aortic dysfunctions [35,36].

In summary, we showed that common and rare variants both contribute to the genetic etiology of CoA, with rare *MYH6* variants and common *PCDHA* delCNVs together accounting for nearly a third of CoA cases. Given atrial dominance of the human and mouse embryonic heart and with *MYH6* and *PCDHA* both being predominantly atrial specific, this suggests the genetic etiology of CoA may be further elucidated by examining other genes regulating atrial contractility in the embryonic heart.

## 6. Limitations

A major limitation in this study is the relatively small cohort size, which could have underestimated the contribution of rare *MYH6* variants in CoA and other LVOTO CHD. This may explain the failure to detect significant association of *MYH6* variants with HLHS. The small sample size also may account for the paucity of CoA cases with co-occurrence of rare *MYH6* variants with the common *PCDHA* delCNVs.

## Figures and Tables

**Figure 1 genes-13-00636-f001:**
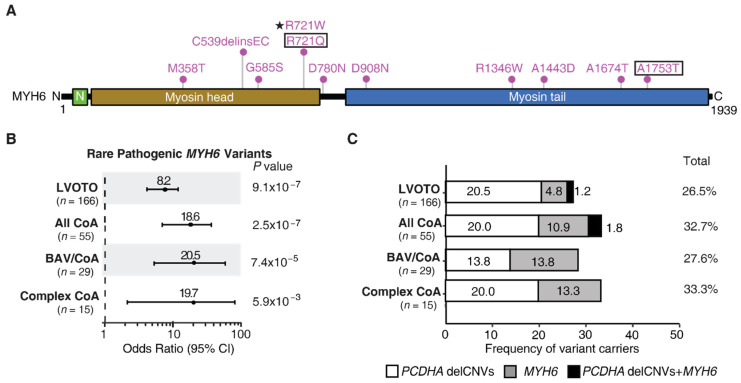
Rare pathogenic *MYH6* variants enriched in LVOTO cohort. (**A**) Pathogenic *MYH6* variants identified in Pittsburgh cohort. Variants R721Q and A1753T (with black box) were found in the same patient. The Icelandic R721W variant is indicated with a star. (**B**) Burden analysis yielded odds ratio indicating significant excess of rare pathogenic *MYH6* variants associated with CoA and other related LVOTO. *p* values were obtained by Fisher’s exact test. 95% CI: 95% confidence interval. (**C**) Proportion of subjects with *PCDHA* delCNVs (16.8 kb/13.6 kb) and/or pathogenic *MYH6* variants among LVOTO in Pittsburgh cohort.

**Figure 2 genes-13-00636-f002:**
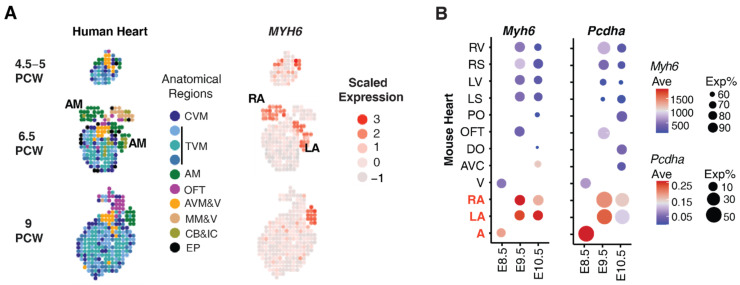
Atrial specific expression of *MYH6* and the *PCDHA* gene cluster. (**A**) Spatial transcriptomic heatmap revealed abundant *MYH6* expression predominantly in the atria of embryonic heart at 4.5–5, 6.5 and 9 postconceptional weeks (PCW). Heart regions color coded as shown. CVM: compact ventricular myocardium. TVM: trabecular ventricular myocardium. AM: atrial myocardium. OFT: outflow tract/large vessels. AVM&V: atrioventricular mesenchyme & valves. MM &V: mediastinal mesenchyme & vessels. CB & IC: cavities with blood and immune cells. EP: epicardium. (**B**) Atria = expression of *Myh6* and *Pcdha* gene cluster in mouse embryonic heart. Dot size shows proportion of cells expressing *Pcdha* (Exp%). Color denotes average expression (Ave). A, atrium; V, ventricle; RA, right atrium; LA, left atrium; AVC, atrioventricular canal; RV, right ventricle; RS, right ventricular septum; LS, left ventricular septum; LV, left ventricle; OFT, outflow tract; PO, proximal outflow tract; DO, distal outflow tract.

**Figure 3 genes-13-00636-f003:**
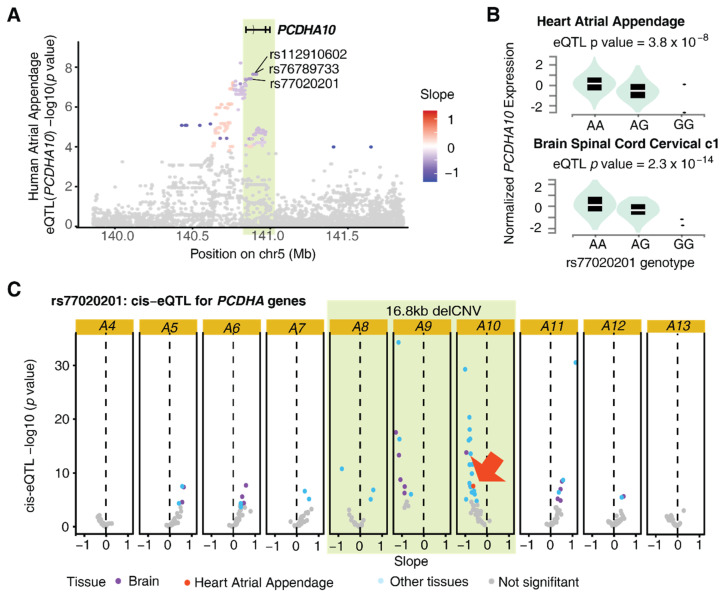
cis-eQTL analysis of PCDHA delCNV impact on *PCDHA* gene expression. (**A**) GTEx database showed in the atria, three tagSNPs among top 5 cis-eQTLs were associated with significant change in *PCDHA10* expression. (**B**) eQTL obtained using GTEx data showed reduced expression *of PCDHA10* associated with the 16.8 kb *PCDHA* delCNV tracked by rs7702020 (A/G, non-risk/risk allele). *p*-values adjusted to the β distribution. (**C**) Volcano plot shows cis-eQTL tracked expression of the *PCDHA* cluster in different human tissue via rs77020201 for the 16.8 kb *PCDHA* delCNV. Negative/positive slope indicates decrease/increase expression respectively. Grey dots denote nonsignificant cis-eQTL associations.

**Figure 4 genes-13-00636-f004:**
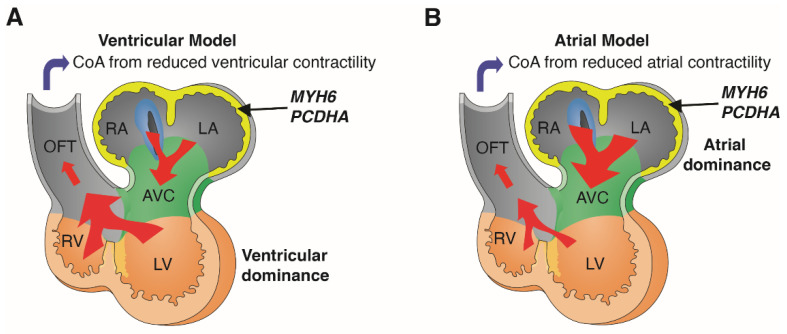
Atrial vs. ventricular hemodynamic model of coarctation. *MYH6* and *PCDHA* are both expressed predominantly in the atria during early human and mouse embryonic development. Ventricular model of coarctation (CoA) (**A**) proposes that flow provided by ventricular contractile force plays a predominant role in the developmental etiology of CoA (**A**). In contrast, in the atrial model of CoA (**B**), flow provided by atrial contraction plays a predominant role in the developmental etiology of CoA. Diagrams adapted with permission from van Eif et al. (2018) [32]. **RA**, right atrium; **LA**, left atrium; **AVC**, atrioventricular canal; **RV**, right ventricle; **LV**, left ventricle; **OFT**, outflow tract.

## Data Availability

The MYH6 variants identified in Pittsburgh cohort are available under SRA numbers BioProject accession number PRJNA632119. Publicly available single-cell RNA sequencing raw data were downloaded from the NCBI/GEO database under accession number GSE76118. Spatial transcript display for MYH6 in Figure 2A was downloaded from the spatial transcriptomics explorer website (https://spatialtranscriptomics3d.shinyapps.io/Developmental_heart_explorer/, accessed on 20 July 2021). GTEx V8 cis-eQTLs data in European-American subjects was directly downloaded from the GTEx website on 27 Sep 2021. Phasing data of the 1000 Genome Project was directly downloaded from 1KG FTP: ftp://ftp.ncbi.nih.gov/1000genomes/ftp/release/20130502/ (accessed on 8 October 2019). The variant dataset of GnomAD exome v2.1.1 was directly downloaded from the GnomAD database: https://gnomad.broadinstitute.org/downloads (accessed on 20 March 2022).

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
