# Peer review of "Rare and Common Variants Uncover the Role of the Atria in Coarctation of the Aorta"

_genes, 2022, doi:10.3390/genes13040636_

Round 1

Reviewer 1 Report

a genetic investigation in patients with aortic coarcation (CoA) is presented in this study. The paper is well presented and interest for the reader. The major concern is that CoA leads to a heamodynamic impairment. Thus, the present study will have a much higher impact if the association between genetic variants and flow disturbances in the CoA are investigated. MRI could be used to assess flow for correlation with biomarkers. The following minor comments needs to be addressed:

I suggest authors can refere the following to support this statement: These are related to biomarkers and related hamodynamic in BAV and are quite related to the present study:

[1]Gallo A, Agnese V, Coronnello C, Raffa GM, Bellavia D, Conaldi PG, Pilato M, Pasta S. On the prospect of serum exosomal miRNA profiling and protein biomarkers for the diagnosis of ascending aortic dilatation in patients with bicuspid and tricuspid aortic valve. Int J Cardiol. 2018 Dec 15;273:230-236. doi: 10.1016/j.ijcard.2018.10.005. Epub 2018 Oct 5. PMID: 30297190.

[2] Pasta S, Agnese V, Gallo A, Cosentino F, Di Giuseppe M, Gentile G, Raffa GM, Maalouf JF, Michelena HI, Bellavia D, Conaldi PG, Pilato M. Shear Stress and Aortic Strain Associations With Biomarkers of Ascending Thoracic Aortic Aneurysm. Ann Thorac Surg. 2020 Nov;110(5):1595-1604. doi: 10.1016/j.athoracsur.2020.03.017. Epub 2020 Apr 11. PMID: 32289298.

page 2 last sentence: ay, have the author access to flow data for echo or MRI to correlate the CoA heamodynamic to the genetic markers?

Page 3 line 5. Please state the level of significance that was used in this study. 

Fig 2, 3 and 4: there is very long caption. Please consider to move part of text in the result or discussion section.

Author Response

We appreciate the reviewer’s comments.

Reviewer 2 Report

The manuscript addresses an interesting topic on MYH6 variant and PCDHA delCNVs are significantly associated with CoA and BAV.

Introduction: The main presentation of severe CoA neonatally is shock rather than cyanosis

Material and methods: suggest to have flow diagram about the steps of genetic investigation performed to facilitate understanding. Figure's caption should be better.

Is there any study done to determine the modes of inheritance for these variants 

Genetic condition that are commonly associated with Bicuspid aortic valve and CoA is Turner syndrome(45X0), is there any study to investigate these variants on this group of patients

Since the severity of CoA will determine the age of presentation, is there any information any association  of this variants with severity of CoA

Results section : should not include objective

Discussion: The statement of  "deficiency of either gene may cause decrease atrial  contractility with reduced blood flow that lead to CoA" may not be accurate since the main function of the atria as a reservoir and conductance. Only about 20% of ventricular filling contributed by atrial contraction. Probably need further investigation is there any role of atrial  in development of CoA

Author Response

We appreciate reviewer’s comments.

Reviewer 3 Report

This manuscript strengthens the current knowledge regarding the initial mechanisms that are involved in the occurrence of the coarctation of the aorta. It provides genetic evidence that supports the role of the atria in the occurrence of the coarctation of the aorta. This is a topic with little evidence and I believe that this work contributes significantly to elucidating the etiopathogenesis of this congenital heart disease. 
The manuscript is well structured. The authors described in detail the methodology and results of the research. The insertion of a graphic image of the concept for which they bring evidence is appreciated. 

Recommendations: 
Revise the title. It seems incomplete.
Revise abbreviation for LVOTO = left ventricular outflow tract obstructions
Please rephrase:  As we also recently also uncovered a role for common variants in two deletion copy number variants in the protocadherin a gene cluster (PCDHA delCNV) in the pathogenesis of LVOTO lesions…  (this sentence lacks clarity) 
Please revise the first sentence in the Discussion. 

Thank you!

Author Response

We appreciate the reviewer’s comments.

Round 2

Reviewer 2 Report

The manuscript was corrected, and comments were addressed appropriately 
The reference 32 and 36 are the same 

Author Response

We appreciate reviewer’s comments. 

Response:  We revised the reference list.